# Omega-3 Fatty Acids Upregulate SIRT1/3, Activate PGC-1α via Deacetylation, and Induce Nrf1 Production in 5/6 Nephrectomy Rat Model

**DOI:** 10.3390/md19040182

**Published:** 2021-03-26

**Authors:** Sung Hyun Son, Su Mi Lee, Mi Hwa Lee, Young Ki Son, Seong Eun Kim, Won Suk An

**Affiliations:** 1Department of Internal Medicine, BHS Han Seo Hospital, Busan 48253, Korea; sohnseung@hanmail.net; 2Department of Internal Medicine, Dong-A University, Busan 49201, Korea; sumilee@dau.ac.kr (S.M.L.); kidney@dau.ac.kr (Y.K.S.); sekim@dau.ac.kr (S.E.K.); 3Department of Anatomy and Cell Biology, Dong-A University, Busan 49201, Korea; hero15p@nate.com

**Keywords:** peroxisome proliferator-activated receptor gamma coactivator-1 alpha, nuclear respiratory factor 1, nuclear factor erythroid 2-related factor 2, sirtuin 1, sirtuin 3, omega-3 fatty acid

## Abstract

Mitochondrial dysfunction contributes to the pathogenesis of kidney injury related with cardiovascular disease. Peroxisome proliferator-activated receptor gamma coactivator-1 alpha (PGC-1α) protects renal tubular cells by upregulating nuclear factor erythroid 2-related factor 2 (Nrf2). AMP-activated protein kinase (pAMPK)-mediated phosphorylation and sirtuin 1/3 (SIRT1/3)-mediated deacetylation are required for PGC-1α activation. In the present study, we aimed to investigate whether omega-3 fatty acids (FAs) regulate the expression of mediators of mitochondrial biogenesis in 5/6 nephrectomy (Nx) rats. Male Sprague-Dawley rats were assigned to the following groups: sham control, Nx, and Nx treated with omega-3 FA. The expression of PGC-1α, phosphorylated PGC-1α (pPGC-1α), acetylated PGC-1α, and factors related to mitochondrial biogenesis was examined through Western blot analysis. Compared to the control group, the expression of PGC-1α, pAMPK, SIRT1/3, Nrf1, mTOR, and Nrf2 was significantly downregulated, and that of Keap 1, acetylated PGC-1α, and FoxO1/3, was significantly upregulated in the Nx group. These changes in protein expression were rescued in the omega-3 FA group. However, the expression of pPGC-1α was similar among the three groups. Omega-3 FAs may involve mitochondrial biogenesis by upregulating Nrf1 and Nrf2. This protective mechanism might be attributed to the increased expression and deacetylation of PGC-1α, which was triggered by SIRT1/3.

## 1. Introduction

Kidneys are involved in regulating several functions, including various metabolic pathways, water and electrolyte balance, blood pressure control, and production of several hormones. As these functions consume enormous energy, kidneys are rich in mitochondria [1]. Mitochondria can adapt to different metabolic conditions via regulating mechanistic target of rapamycin (mTOR) and AMP-activated protein kinase (AMPK)-mediated nutrient sensing pathways to maintain healthy mitochondrial population [2].

Among the studies on mitochondria, to date, there are only a few studies on mitochondrial biogenesis. Most studies related to mitochondrial biogenesis are based on non-renal tissue [3,4], and studies on kidneys are mainly related to acute kidney injury (AKI) animal models, proximal tubular cells, and diabetic nephropathy [5,6,7]. Although mitochondrial biogenesis may link to pathogenesis of mitochondrial dysfunction in non-diabetic chronic kidney disease (CKD), research on this topic is also very limited. CKD and mitochondrial biogenesis are related to cardiovascular disease [8,9]. Improving mitochondrial biogenesis may be beneficial for cardioprotection in CKD.

A study indicating that omega-3 fatty acids (FAs) facilitate the biogenesis of mitochondria was conducted using non-kidney tissues [3,10]. A recent study demonstrated a promising role of omega-3 FAs in mitochondrial biogenesis in animal models with cardiovascular and neurodegenerative diseases [11]. Therefore, in the present study, we aimed to investigate mitochondrial biogenesis in kidneys of 5/6 nephrectomy (Nx) rats. Additionally, we evaluated the effect of omega-3 FAs on mitochondrial biogenesis using this model.

## 2. Results

### 2.1. Changes in Kidney Function and Histological Findings

Compared to the control group, blood urea nitrogen (BUN) level in the Nx and Nx treated with omega-3 FA groups were significantly increased (control group 17.7 ± 1.5, Nx group 77.7 ± 28.4, and Nx with omega-3 FA group 63.9 ± 17.0 mg/dL, *p* = 0.007). Although there was no significant difference between the Nx and Nx treated with omega-3 FA groups, BUN levels were numerically lower in the Nx treated with omega-3 FA group. The serum creatinine (sCr) level among the three groups was similar to that of BUN (control group, 0.4 ± 0.0; Nx group, 1.3 ± 0.6; omega-3 FA, 1.0 ± 0.3 mg/dL; *p* = 0.008). Compared to the control group, severe tubular dilatation and atrophy were observed using PAS staining in the Nx group (Appendix A). Furthermore, the results showed that the Nx treated with omega-3 FA group exhibited less tubulointerstitial changes than the Nx group. The renal function and histological changes in the three groups have already been reported in a previous study [12].

### 2.2. Changes in Factors Related to Mitochondrial Biogenesis

#### 2.2.1. Nrf1 and Nrf2 Expression

Compared to the control group, the Nx group showed significantly decreased expression level of nuclear respiratory factor 1 (Nrf1) and nuclear factor erythroid 2-related factor 2 (Nrf2). However, in the Nx treated with omega-3 FA group, the expression level of Nrf1 and Nrf2 was significantly increased compared to the Nx group, but did not reach the level of control group (Figure 1, Appendix A). These results were also found in Nrf1 mRNA quantitative real-time PCR analysis (Appendix A).

#### 2.2.2. Changes in PGC-1α Expression and Its Activity

The expression of peroxisome proliferator-activated receptor gamma coactivator 1-alpha (PGC-1α) in the Nx group was significantly decreased compared to the control group. The Nx treated with omega-3 FA group showed significantly increased PGC-1α expression compared to the Nx group, but it did not reach the level of control group (Figure 2A, Appendix A). These results were also found in PGC-1α mRNA quantitative real-time PCR analysis (Appendix A). To assess PGC-1α activity, we evaluated the phosphorylation status (the ratio of phosphorylated PGC (pPGC)-1α to PGC-1α) and acetylation status (the ratio of acetylated PGC-1α to PGC-1α) of PGC-1α. The phosphorylation status of PGC-1α in the Nx and Nx treated with omega-3 FA groups was not significantly different compared to the control group. However, the acetylation status of PGC-1α was significantly increased only in the Nx group (Figure 2B). Moreover, the deacetylation status of PGC-1α was significantly decreased only in the Nx group, but in the Nx treated with omega-3 FA group, it was not different from that in the control group.

#### 2.2.3. Expression of Factors Related to PGC-1α Activity: pAMPK, SIRT1/3

Compared to the control group, the Nx and Nx treated with omega-3 groups exhibited significantly lower levels of phosphorylated AMPK (the ratio of pAMPK to AMPK) (Figure 3A). Moreover, the Nx group exhibited significantly lower level of sirtuin 1 (SIRT1) expression compared to the control group (Figure 3B, Appendix A). However, following the addition of omega-3 FA, SIRT1 expression was relatively increased, but this difference was not statistically significant. These results were found in SIRT1 mRNA quantitative real-time PCR analysis (Appendix A). Conversely, sirtuin 3 (SIRT3) expression was significantly decreased in the Nx group compared to the control group, which was rescued significantly in the Nx treated with the omega-3 FA group (Figure 3C).

#### 2.2.4. Keap1, mTOR, FoxO1, and FoxO3 Expression

Compared to the control group, the expression of Kelch-like ECH-associated protein 1 (Keap1) and Forkhead box proteins O1 and O3 (FoxO1/3) was significantly increased in the Nx group. In the Nx treated with omega-3 FA group, the expression level of Keap1 and FoxO1/3 was significantly decreased compared to the Nx group (Figure 4C,E,F). Compared to the control group, the expression of mTOR was significantly decreased in the Nx group. However, in the Nx treated with omega-3 FA group, mTOR expression was rescued, but did not reach the level in the control group (Figure 4B).

#### 2.2.5. Content of Mitochondrial DNA (mtDNA)

A significant reduction of mtDNA was observed in the Nx group compared to control group. However, the reduction amount of mtDNA was significantly recovered in the Nx-treated omega-3 Nx group (Appendix A).

## 3. Discussion

PGC-1α, which is activated via phosphorylation and deacetylation, is the master regulator of mitochondrial biogenesis and energy production. We found a decrease in the expression and deacetylation of PGC-1α in CKD model. It is noteworthy that decreased PGC-1α deacetylation is the main mechanism contributing to dysfunctional mitochondrial biogenesis and reduced energy production in CKD. AMPK activates PGC-1α via phosphorylating its threonine or serine residues [13,14]. AMPK is also activated upon phosphorylation when the AMP/ATP ratio is high. However, AMPK is known to be inactivated in CKD due to the impaired sensing of high AMP levels [15,16]. Similar with previous studies, we demonstrated that AMPK phosphorylation was significantly decreased in kidney tissue of the Nx group. Although pPGC-1/PGC-1α appears to be increased to a certain extent in the Nx group, it is unlikely that the level of PGC-1α phosphorylation is increased because of decreased PGC-1α expression. Moreover, the Nx treated with omega-3 FA group also showed similar results as that of the Nx group, which indicates that omega-3 FA administration did not affect the phosphorylation of AMPK and PGC-1α. Deacetylation can occur in the entire sequence of PGC-1α, and SIRT1 and 3 play a major role in the process of deacetylation [13,14]. After omega-3 FA administration, the expression level of PGC-1α, SIRT1, and 3 increased, resulting in the rescue of deacetylated PGC-1α to the same level as that of the normal control group. The rescue in the expression level of Nrf1 and Nrf2, which is demonstrated in the Nx treated with omega-3 FA group, is directly related to the increase in SIRT1 and 3 expression level, which further led to the deacetylation of PGC-1α.

Recent studies on mitochondria in kidney disease have shown that the increase in PGC-1α or SIRT expression level through genetic or pharmacological interventions improved kidney damage in AKI animal models [5,6,7,17,18]. The underlying mechanism involves the effect of improvement in FA oxidation or antioxidative response. A previous study reported that omega-3 FA administration improved mitochondrial FA beta-oxidation and exhibited antioxidant effects in a 5/6 Nx rat model [19]. There are also reports suggesting that omega-3 FA increases the expression of PGC-1α, Nrf1, and SIRT1 in muscle cells, macrophages, and nerve cells [3,10,11]. As the effects of omega-3 FAs on mitochondria in a CKD model have not been reported yet, we demonstrated that omega-3 FA is effective in recovering mitochondrial biogenesis-related molecules and mtDNA using a CKD model in this study.

Both mTOR and AMPK are involved in mitochondrial biogenesis, but their roles vary depending on the nutritional status. mTOR, a target of rapamycin complex 1 (mTOC1), can be activated via energy stimuli, such as amino acid or glucose, and can trigger pathways that lead to anabolic processes and mitochondrial biogenesis. Conversely, AMPK can be activated via hypoxia and nutritional deficit, which leads to the activation of catabolic process and mitochondrial biogenesis, and inhibits energy consumption through mTOC1 [2]. In this study, we not only demonstrated a decrease in pAMPK/AMPK, but also a decrease in mTOR in the CKD model. Furthermore, omega-3 FA administration led to a near recovery of mTOR expression but no recovery of the decrease in pAMPK/AMPK. Therefore, omega-3 FAs might alleviate decreased mitochondrial biogenesis by increasing mTOR expression in kidneys of a CKD animal model. However, further studies are required to be conducted to clarify whether mTOR is decreased in CKD patients, and how the increase in mTOR expression through omega-3 FA treatment will affect the mitochondrial health and CKD progression.

FoxO1 and 3 bind to PGC-1α promoter and enhance its transcription [20,21]. However, in this study, FoxO1 and 3 were found to be increased in the Nx group and decreased in the control and Nx treated with omega-3 FA groups. Therefore, it is assumed that the expression of FoxO1 and 3 is relevant for the compensatory response to decrease PGC-1α in CKD or increase PGC-1α through omega-3 FA administration. Nrf2 plays an important role in cellular redox homeostasis and mitochondrial biogenesis. There are three ubiquitin ligase systems related to the Nrf2 degradation (Keap1, β-transducin repeats-containing protein, and E3 ubiquitin ligase synoviolin) [22]. In this study, we showed a significant decrease in Nrf2 and a significant increase in Keap1 in the Nx group compared to the control group.d However, in the Nx treated with omega-3 FA group, the expression of Keap1 was found to be decreased and Nrf2 was increased. The results suggest that the improved antioxidative effect of omega-3 FA administration in the CKD rat model may be related to the decrease in Keap1 expression and Nrf2 degradation, which might enhance mitochondrial biogenesis. Based on the results obtained in this study, further research is required to elucidate the causal relationship between FoxO1/3 and Nrf2 in the CKD environment, and the effect of omega-3 FA administration.

The molecular weight of Nrf2 was found at 68  kilodalton (kDa) area in this study and other recent studies [23,24]. However, previous studies indicated biologically relevant molecular weight of Nrf2 is 95–110 kDa with evidence [25,26]. We also found bands which were suspected as biologically relevant Nrf2 (Appendix A). Further studies are necessary to find the molecular weight of biologically important Nrf2.

In previous studies, the changes in mitochondria on the 28th day after 5/6 Nx have been reported, which indicated a decrease in gene or protein expression associated with beta oxidation, oxidative phosphorylation, and inner membrane structural protein, but there was no result related to PGC-1 expression [27,28]. In this study, we confirmed that mitochondrial biogenesis-related molecules and mtDNA reflecting mitochondrial biogenesis were modulated during the chronic stage, as the changes were observed on the 45th day after 5/6 Nx in rats. Despite the histological and biological improvements, the omega-3 FA treatment did not result in functional recovery of kidneys compared to the Nx model. Therefore, further research is required to determine whether omega-3 FAs are effective in restoring kidney function by inducing effects that might improve mitochondrial function.

There are several limitations in this this study. First, we did not evaluate citrate synthase activity and superoxide formation in the mitochondrial respiration chain. Second, we did not experiment which FA components of Omacor are mainly affected on mitochondrial biogenesis-related molecules. Third, we did not demonstrate a clear mechanism for improving mitochondrial biogenesis-related molecules. We just suggest that anti-inflammatory effect, reducing oxidative stress and omega-3 fatty acid itself as metabolic compounds, may be related with the mechanism.

## 4. Materials and Methods Mitochondrial Biogenesis

### 4.1. Animals and Experimental Design

Studies were conducted on male Sprague-Dawley rats (9-weeks-old) purchased from Japan SLC, Inc. (Shizuoka, Japan). All rats were subjected to either sham control or 5/6 Nx. All procedures involving animals were performed in accordance with approval of Dong-A University’s Institutional Animal Care Committee (DIACUC-14-4). Rats were kept under standard condition and allowed free access to tap water and standard diet.

All rats were assigned to the following groups: group 1 (sham control, *n* = 6), control rats kept on saline (1 mL/kg/day by gastric lavage); group 2 (Nx group, *n* = 6), Nx rats kept on saline (1 mL/kg/day by gastric lavage); group 3 (omega-3 FA group), Nx rats treated with omega-3 FAs. The dose and administration route of omega-3 FAs (Omacor, 300 mg/kg/day by gastric gavage, Pronova Biocare, Sandefjord, Norway) were determined based on previous studies [19]. Omacor is marine-originated omega-3 FA and is made up of 460 mg eicosapentaenoic acid (EPA) and 380 mg of docosahexaenoic acid (DHA) in 1 g of Omacor. Omacor is usually prescribed for secondary prevention after myocardial infarction and hypertriglyceridemia treatment [29]. Two omega-3 FAs (EPA plus DHA) are major (84%) and active components of Omacor, which are pharmaceutically prepared compositions overcoming minor components (16%). We chose Omacor for omega-3 FA supplementation with anti-inflammatory and reducing oxidative stress, because it decreased the risk of cardiovascular disease in the clinical studies [30]. The rats were fed evenly and their weight was observed daily. Rats were allowed free access to tap water for 15 weeks after surgery, and then euthanized under diethyl ether anesthesia. At death, an aortic blood sample was collected and then centrifuged at 3000 rpm for 10 min. To identify changes in kidney function, BUN and sCr levels were measured using an automatic analyzer (Roche, Germany).

### 4.2. Histopathologic Examination

Formalin-fixed kidney tissues were embedded in paraffin, cut into slices (4 μm), and stained with periodic acid-Schiff. Histopathologic changes were observed with Aperio ScanScope (Aperio Technologies, Vista, CA, USA).

### 4.3. RNA Isolation and Quantitative Real-Time PCR Analysis

Total RNA was extracted from kidney tissues using TRIzol reagent. Then, 1 μg of total RNA was converted into single-stranded cDNA using M-MLV cDNA synthesis kit (Enzynomics, Daejeon, Korea). Primers were designed from respective gene sequences using Primer3 and mfold software. For quantitative real-time PCR analysis, cDNA was subjected to PCR amplification using gene-specific primers: rat PGC1α 5′- CCG AGA ATT CAT GGA GCA AT-3′ (sense), 5′- GTG TGA GGA GGG TCA TCG TT-3′ (antisense); rat Nrf1 5′- GTT TCA TGG ACC CAA GCA TT-3′ (sense), 5′- GGT GGC CTG AGT TTG TGT CT-3′ (antisense); rat SIRT3, 5′- GAG ACT TGG TGG GGT CCT TT -3′ (sense), 5′- ATC CTG CAG CTC TTG TGT CC -3′ (antisense); rat SIRT1, 5′- CAG GTT GCA GGA ATC CAA AG -3′ (sense), 5′- CTC CAC GAA CAG CTT CAC AA -3′ (antisense); rat β-actin, 5′- GCG CAA GTA CTC TGT GTG GA -3′ (sense), 5′- CAT CGT ACT CCT GCT TGC TG -3′ (antisense). Real-time PCR was performed using SYBR Green PCR Master Mix (Applied Biosystems, Foster City, CA, USA) with an ABI 7500 instrument (Applied Biosystems, Waltham, MA, USA).

### 4.4. Western Blotting and Immunoprecipitation

Western blotting was analyzed as described previously with slight modifications [10]. Kidney tissues were homogenized by lysis buffer (PRO-PREP protein extraction solution), incubated (30 min at 4 °C), and centrifuged (14,000 rpm for 20 min at 4 °C). The concentration of protein was determined by Bradford protein assay reagent (Bio-Rad, Hercules, CA, USA). Equal protein sample was loaded on a 7.5–15% SDS-PAGE and transferred to a nitrocellulose membrane (Amersham Pharmacia Biotech, Piscataway, NJ, USA). Then, proteins were immunoblotted with each antibody. Antibodies against PGC-1α and SIRT1 were obtained from Santa Cruz Biotechnology (Santa Cruz, CA, USA). Antibodies against AMPK, pAMPK, Nrf1, SIRT3, FoxO1, FoxO3a, and mTOR were purchased from Cell Signaling Technology (Danvers, MA, USA). Antibodies against Nrf2, Keap1 (monomer), and β-actin were purchased from Abcam (Cambridge, MA, USA) and Sigma (St. Louis, MO, USA). For immunoprecipitation, a total of 500 μg protein was incubated with PGC-1α antibodies (Santa Cruz Biotechnology, Santa Cruz, CA, USA) and protein A/G plus-agarose beads (Santa Cruz Biotechnology, Santa Cruz, CA, USA), and allowed to mix at 4 °C overnight. The immunoprecipitates were then separated using SDS-PAG electrophoresis and immunoblotted using the acetyl-lysine (Cell Signaling Technology, Beverly, MA, USA) and phosphoserine (MerckMillipore, Burlington, MA, USA) antibodies. The membranes were subsequently incubated with horseradish peroxidase-conjugated secondary antibody for 60 min at room temperature. Protein levels were standardized by β-actin (ImageJ version 1.48q). Western blotting and immunoprecipitation analysis with antibodies was performed using the Super Signal West Pico enhanced chemiluminescence substrate (Thermo Scientific, Hudson, NH, USA) and detected using LAS-3000 Plus (Fuji Photo Film, Tokyo, Japan).

### 4.5. Measurement of mtDNA Content

The qPCR was used to determine relative content of mtDNA. Reaction was performed via SYBR Green chemistry using 3 ng of total DNA as a template and the following primers: rat mtDNA 5′-GGTTCTTACTTCAGGGCCATCA-3′ (sense), 5′-TGATTAGACCCGTTACCATCGA-3′ (antisense); rat β-actin, 5′- CCCAGCCATGTACGTAGCCA -3′ (sense), 5′- CGTCTCCGGAGTCCATCAC -3′ (antisense). The mtDNA content relative to the nuclear DNA was reported in [31].

### 4.6. Statistical Analysis

Statistical analyses were performed using the SPSS 18.0 software (IBM Corp., Armonk, NY, USA). Results are expressed as mean ± SD. The means for the groups were compared by analysis of variance followed by Tukey’s multiple comparison. *p* < 0.05 was considered statistically significant.

## 5. Conclusions

Conclusively, significant modulations in the expression of mediators related to mitochondrial biogenesis were observed in a CKD rat model. Omega-3 FA may improve mitochondrial biogenesis by upregulating Nrf1 and Nrf2. This protective mechanism may be initiated through an increase in PGC-1α expression and deacetylation of PGC-1α, which was triggered by increased SIRT1/3 production (Figure 5).

## Figures and Tables

**Figure 1 marinedrugs-19-00182-f001:**
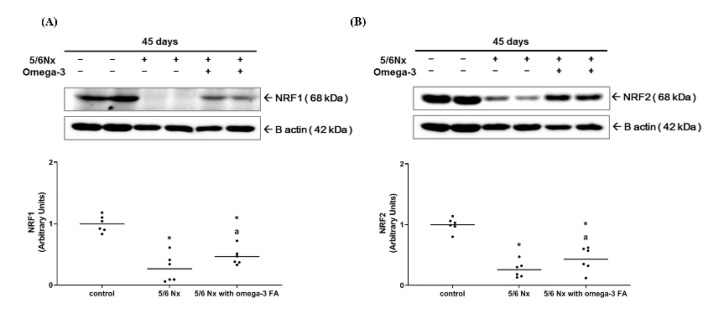
Expressions of (**A**) Nrf1 and (**B**) Nrf2 (*n* = 6/group). Decreased Nrf 1 and Nrf2 expressions of 5/6 nephrectomy group were improved by omega-3 FA supplementation. * *p* value < 0.05 (mean values are significantly different from control). ^a^
*p* value < 0.05 (mean values are significantly different from the Nx group).

**Figure 2 marinedrugs-19-00182-f002:**
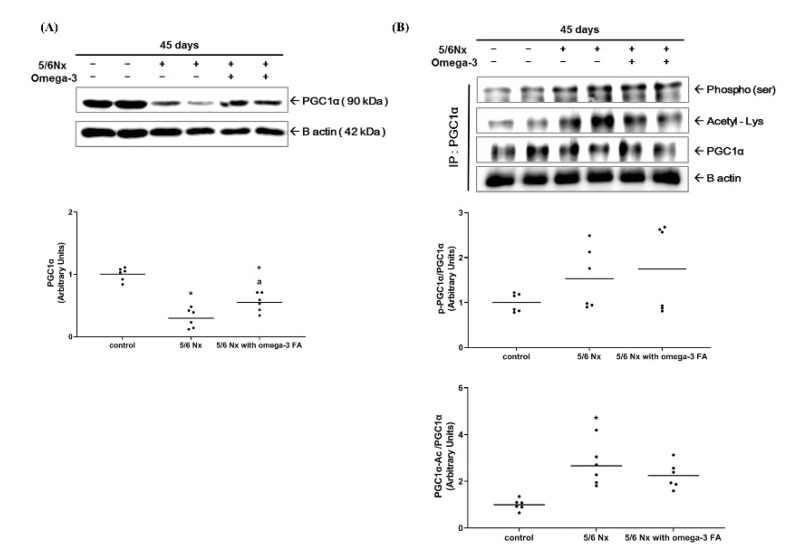
Changes in the (**A**) expression of PGC-1α and (**B**) activity of PGC-1α (*n* = 6/group). Decreased PGC-1α expressions of 5/6 nephrectomy group were improved by omega-3 FA supplementation. The deacetylation status of PGC-1α was improved by omega-3 FA supplementation. p-PGC-1α, phosphorylated PGC-1α. PGC-1α-Ac, acetylated PGC-1α. * *p* value < 0.05 (mean values are significantly different from control). ^a^
*p* value < 0.05 (mean values are significantly different from the Nx group).

**Figure 3 marinedrugs-19-00182-f003:**
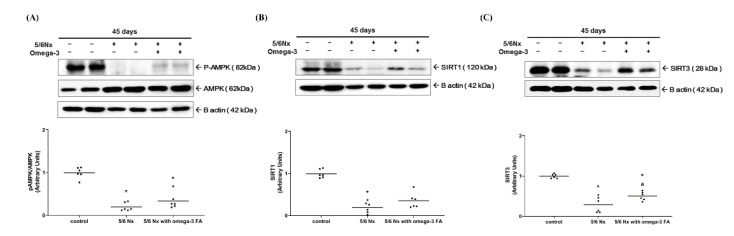
Changes in the expression of factors related to the function of PGC-1α (*n* = 6/group): (**A**) phosphorylation of AMPK (the ratio of pAMPK to AMPK), (**B**) expression of SIRT1, and (**C**) SIRT3. Decreased SIRT3 expressions of 5/6 nephrectomy group were improved by omega-3 FA supplementation. * *p* value < 0.05 (mean values are significantly different from control). ^a^
*p* value < 0.05 (mean values are significantly different from the Nx group).

**Figure 4 marinedrugs-19-00182-f004:**
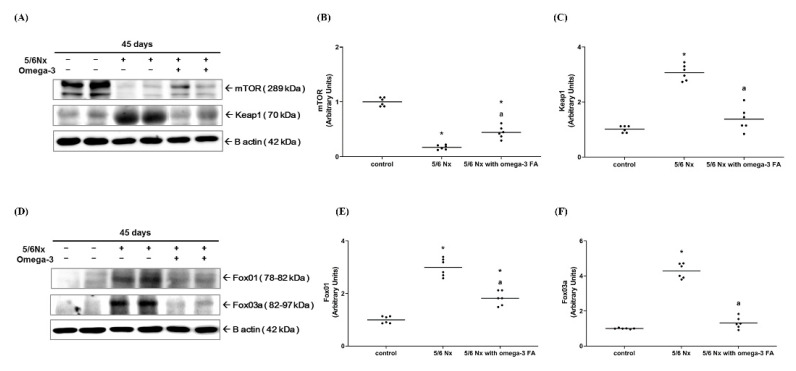
Changes in the expression of mTOR and Keap1 (**A**–**C**, *n* = 6/group). Changes in the expression of FoxO1 and FoxO3 (**D**–**F**, *n* = 6/group). Decreased mTOR and increased Keap1, FoxO1, and FoxO3 expressions of 5/6 nephrectomy group were improved by omega-3 FA supplementation. * *p* value < 0.05 (mean values are significantly different from control). ^a^
*p* value < 0.05 (mean values are significantly different from the Nx group).

**Figure 5 marinedrugs-19-00182-f005:**
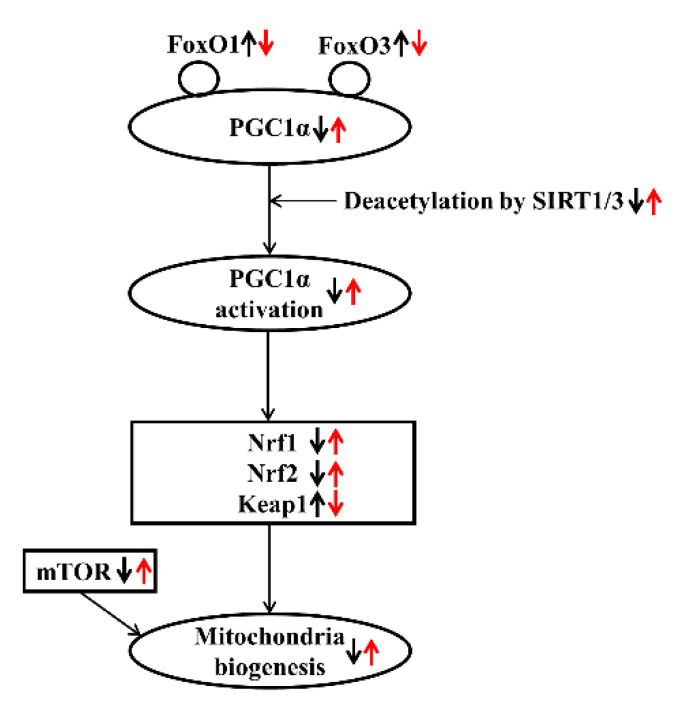
Effect of omega-3 FA on mitochondrial biogenesis in CKD. Bold black arrows indicate the expression of mediators related to mitochondrial biogenesis in a CKD rat model. Red arrows indicate the expression of mediators after omega-3 FA administration. Up/down arrows indicate an increase and decrease in the expression of mediators.

## Data Availability

The data presented in this study are available on request from the corresponding author.

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
