# Peer review of "Omega-3 Fatty Acids Upregulate SIRT1/3, Activate PGC-1α via Deacetylation, and Induce Nrf1 Production in 5/6 Nephrectomy Rat Model"

_marinedrugs, 2021, doi:10.3390/md19040182_

Round 1

Reviewer 1 Report

No further comments, the authors have in the most part addressed my concerns.

Only one minor change, check the formatting on Supplementary Figure 2 the text is misaligned.

Reviewer 2 Report

additional confirmatory work has been performed.

This manuscript is a resubmission of an earlier submission. The following is a list of the peer review reports and author responses from that submission.

Round 1

Reviewer 1 Report

Son and colleagues characterised the effects of a formulation containing omega-3 fatty acids in a rat nephrectomy model. Nrf1, Nrf2, (acetylated) Pgc1alpha, (phosporylated) Ampk, Sirt1, Sirt3, Keap1, Foxo1, Foxo3a, and Mtor expression was determined by immunoblotting. Changes are discussed and hypotheses generated. 1. Which omega-3 FAs constitute the Omacor preparation? Which other compunds are included? I think it is of paramount importance to extensively analyse the mixture and to perform additional experiments to identify the compound(s) that elicits the effect before drawing any conclusions. 2. The mechanism is unclear. Once the compounds responsible are identified, the mechanism could be probed with e.g. etomoxir, PPAR agonists and inverse agonists, antioxidants, etc. to narrow down the target(s). Minor points: Please adhere to the convention for gene and protein names: Rat proteins should be spelled with only the first letter as a capital. The manuscript is inconsistent regarding this (e.g. NRF1 vs. Nrf2). The figure legends are very short and do not include enough information necessary to interpret the data. In my opinion, the study does not include appropriate controls, and hypotheses presented in the model should be tested mechanistically. What is the chain of events elicited by the fatty acid mixture?

Author Response

Son and colleagues characterised the effects of a formulation containing omega-3 fatty acids in a rat nephrectomy model. Nrf1, Nrf2, (acetylated) Pgc1alpha, (phosporylated) Ampk, Sirt1, Sirt3, Keap1, Foxo1, Foxo3a, and Mtor expression was determined by immunoblotting. Changes are discussed and hypotheses generated.

  1. Which omega-3 FAs constitute the Omacor preparation? Which other compunds are included? I think it is of paramount importance to extensively analyse the mixture and to perform additional experiments to identify the compound(s) that elicits the effect before drawing any conclusions.

à Response: Thank you very much for your comments. In accordance with your advice, we included your points in the revised manuscript. Omacor is marine-originated omega-3 FA and is made up 460 mg of eicosapentaenoic acid and 380 mg of docosahexaenoic acid in 1 g of Omacor.

  1. The mechanism is unclear. Once the compounds responsible are identified, the mechanism could be probed with e.g. etomoxir, PPAR agonists and inverse agonists, antioxidants, etc. to narrow down the target(s).

à Response: The authors appreciate the reviewer’s comments. The mechanism of omega-3 fatty acid improving mitochondrial biogenesis related molecules is unclear. Authors added this limitation of study in the revised manuscript as “Second, the mechanism is not clear for improving mitochondrial biogenesis related molecules. We just suggests that anti-inflammatory effect, anti-oxidative effect and omega-3 fatty acid itself as metabolic compounds may be related mechanism.

Minor points: Please adhere to the convention for gene and protein names: Rat proteins should be spelled with only the first letter as a capital. The manuscript is inconsistent regarding this (e.g. NRF1 vs. Nrf2).

à Response: Thank you very much for your comments. Authors corrected it in the revised manuscript.

The figure legends are very short and do not include enough information necessary to interpret the data.

à Response: Thank you very much for your comments. Authors corrected it in the revised manuscript.

In my opinion, the study does not include appropriate controls, and hypotheses presented in the model should be tested mechanistically. What is the chain of events elicited by the fatty acid mixture?

à Response: The authors appreciate the reviewer’s comments. The chain of events elicited by Omacor is improving mitochondrial biogenesis related molecules although the mechanism is unclear. As we mentioned before, we add this limitation in the revised manuscript.

Reviewer 2 Report

The present study aimed to investigate mitochondrial biogenesis in kidneys of 5/6 nephrectomy (Nx) rats. The study evaluates the effect of omega-3 FAs on mitochondrial biogenesis using this model. This is a new follow-up study on results that were described two years ago but it deals with a different subject.

They find that the expression of PGC-1α, pAMPK, SIRT1/3, NRF1, mTOR, and Nrf2 was downregulated whereas Keap 1, acetylated PGC-1α, and FoxO1/3, was significantly upregulated in their rat model. The Omega-3 fatty acid-treated group showed rescue of this protein expression. The work hypothesizes that FAs act on mitochondrial biogenesis by upregulating NRF1 and Nrf2.

This represents an interesting study and directs towards applicability in disease models such as acute/chronic kidney injury.

Author Response

The present study aimed to investigate mitochondrial biogenesis in kidneys of 5/6 nephrectomy (Nx) rats. The study evaluates the effect of omega-3 FAs on mitochondrial biogenesis using this model. This is a new follow-up study on results that were described two years ago but it deals with a different subject.

They find that the expression of PGC-1α, pAMPK, SIRT1/3, NRF1, mTOR, and Nrf2 was downregulated whereas Keap 1, acetylated PGC-1α, and FoxO1/3, was significantly upregulated in their rat model. The Omega-3 fatty acid-treated group showed rescue of this protein expression. The work hypothesizes that FAs act on mitochondrial biogenesis by upregulating NRF1 and Nrf2.

This represents an interesting study and directs towards applicability in disease models such as acute/chronic kidney injury.

à Response: The authors appreciate the reviewer’s comments.

Reviewer 3 Report

This is a generally well thought out paper based on good hypothesis it is just missing key mitochondrial biogenesis data, which would significantly add to the impact of the paper and would test the authors hypothesis further.

Major revisions required before resubmission

  1. Mitochondrial biogenesis assays -/+ Omega 3 supplementation. So for example, mitochondrial mass assays, mtDNA, immunofluorescence staining, mitochondrial respiration or citrate synthase assays.
  2. Are these changes you say post translational or at the mRNA level? QPCR should be performed to answer this.
  3. Changes in kidney function and histological findings should be fully presented showing all data for all n=6
  4. All data should be presented showing individual data points in each graph not just block histograms
  5. The pgc-1 alpha densitometry in Figure 2does not match the picture for the omega 3 treated group, please show the individual data points.
  6. Figure 3-The actin blots look very similar to the blots in Figure 1. Please confirm whether these are the same blots or different. Ideally, the full length gels should be submitted as supplementary data
  7. Figure 4 the actin is the same for A and D, please confirm whether all of these examples were taken from the same blot.
  8. Please add molecular  weight markers to all blots
  9. Figure 4-which form of Keap 1 is this? the monomer or the dimer?-Which is why molecular weight markers should be added

Additional revisions

  1. In all Figure legends please state which specific stats tests were used in this figure
  2. Please state why the specific stats analysis was performed and whether it was based on normality distribution testing (NDT). Please state which methods were used for NDT

Author Response

This is a generally well thought out paper based on good hypothesis it is just missing key mitochondrial biogenesis data, which would significantly add to the impact of the paper and would test the authors hypothesis further.

Major revisions required before resubmission

  1. Mitochondrial biogenesis assays -/+ Omega 3 supplementation. So for example, mitochondrial mass assays, mtDNA, immunofluorescence staining, mitochondrial respiration or citrate synthase assays.

à Response: The authors appreciate the reviewer’s comments. We are very sorry that we cannot add mitochondrial mass assays, mtDNA, immunofluorescence staining, mitochondrial respiration or citrate synthase assays in the revised manuscript. We added these limitations in the revised manuscript; “There are several limitations in this this study. First, we did not evaluate mitochondrial mass assays, mitochondrial DNA, immunofluorescence staining, mitochondrial respiration or citrate synthase assays. Second, …”

  1. Are these changes you say post translational or at the mRNA level? QPCR should be performed to answer this.

à Response: The authors appreciate the reviewer’s comments. Authors added QPCR in the revised manuscript as supplementary data.

  1. Changes in kidney function and histological findings should be fully presented showing all data for all n=6

à Response: Thank you very much for your comments. Authors added it in the revised manuscript as supplementary data. We added 6 histologic findings representing kidney function.

  1. All data should be presented showing individual data points in each graph not just block histograms

à Response: Thank you very much for your comments. Authors corrected it in the revised manuscript.

  1. The pgc-1 alpha densitometry in Figure 2 does not match the picture for the omega 3 treated group, please show the individual data points.

à Response: Thank you very much for your comments. Figure 2 is divided into expression by western blotting (A) and activity by immunoprecipitation (B) of PGC-1α. Authors corrected it in the revised manuscript.

  1. Figure 3-The actin blots look very similar to the blots in Figure 1. Please confirm whether these are the same blots or different. Ideally, the full length gels should be submitted as supplementary data

à Response: Thank you very much for your comments. Authors corrected them in the revised manuscript.

  1. Figure 4 the actin is the same for A and D, please confirm whether all of these examples were taken from the same blot.

à Response: Thank you very much for your comments. Authors corrected it in the revised manuscript.

  1. Please add molecular weight markers to all blots

à Response: Thank you very much for your comments. Authors modified it in the revised manuscript.

  1. Figure 4-which form of Keap1 is this? the monomer or the dimer?-Which is why molecular weight markers should be added

à Response: Thank you very much for your comments. The form of Keap1 is monomer. Authors modified it in the revised manuscript.

Additional revisions

  1. In all Figure legends please state which specific stats tests were used in this figure.

à Response: Thank you very much for your comments. Authors added it in the revised manuscript.

  1. Please state why the specific stats analysis was performed and whether it was based on normality distribution testing (NDT). Please state which methods were used for NDT.

à Response: Thank you very much for your comments. Mann-Whitney U test and Kruskal-Wallis test were used because six sample sizes were not adequate considered as normally distributed. Authors added this point in the revised manuscript.

Round 2

Reviewer 1 Report

Both major points were not addressed. In my opinion, mechanistic insight and, most importantly, clarity regarding the active compound(s) in Omacor are necessary. Omacor is not only composed of the two omega-3 FAs specified in the manuscript. Additional experiments are needed in order to draw any conclusions from the data.

Author Response

Reviewer (#1)' Comments to Author:

Both major points were not addressed. In my opinion, mechanistic insight and, most importantly, clarity regarding the active compound(s) in Omacor are necessary. Omacor is not only composed of the two omega-3 FAs specified in the manuscript. Additional experiments are needed in order to draw any conclusions from the data.

à Response: The authors appreciate the reviewer’s comments. We are sorry that we did not give clear explanations in the 1st revised manuscript. Omacor is usually prescribed for secondary prevention after myocardial infarction and hypertriglyceridemia treatment [1]. Two omega-3 fatty acids (EPA plus DHA) are major (84%) and active components of Omacor, which are pharmaceutically prepared compositions overcoming minor components (16%). We chose Omacor for omega-3 FA supplementation with anti-inflammatory and reducing oxidative stress because it decreased risk of cardiovascular disease in the clinical studies [2]. We added these explanations in the 2nd revised manuscript. We are very sorry that we cannot add additional experiments. We added this limitation of this study in the 2nd revised manuscript.

References

  1. Yusof, H.M.; Cawood, A.L.; Ding, R.; Williams, J.A.; Napper, F.L.; Shearman, C.P.; Grimble, R.F.; Payne, S.P.; Calder, P.C. Limited impact of 2 g/day omega-3 fatty acid ethyl esters (Omacor(R)) on plasma lipids and inflammatory markers in patients awaiting carotid endarterectomy. Mar Drugs 2013, 11, 3569-3581, doi:10.3390/md11093569.
  2. Innes, J.K.; Calder, P.C. Marine Omega-3 (N-3) Fatty Acids for Cardiovascular Health: An Update for 2020. Int J Mol Sci 2020, 21, doi:10.3390/ijms21041362.

Reviewer 3 Report

The authors have addressed the majority of comments including adding additional data. However they have not addressed my first and most important issue with the paper which is lack of any mitochondrial biogenesis assay. As previously stated this would add real weight to their hypothesis. The authors have stated this cannot be performed and have added a limitations section to offset this lack of experimental input. 

If the manuscript is to be accepted it still needs further information around which distribution tests were performed on the data and why certain statistical tests were performed . A Shapiro-Wilk test and Kolmogorov-Smirnov test can be performed on data sets of six. So I dont accept the authors current explanation of their use of non-parametric testing. Moreover, the authors state they use Mann-Whitney U test and Kruskal-Wallis tests, but every Figure legend states that Mann-Whitney U tests were used. They need to justify why they are using non-parametric t-tests and only comparing two groups instead of using Kruskal wallis and comparing all three groups

The authors state when describing their SIRT3 mRNA data that

"changes were not found in SIRT3 mRNA quantitative real-time PCR analysis because one SIRT3 mRNA expression was abnormally higher in Nx group (Supplementary Figure 4)" 

This is true but there is also two distinct populations in their controls which needs to be commented on and an explanation offered.

The authors have altered the actin blots as requested however the original targets blots have remained the same. So where have these actin blots come from? This should be made clear to the Editor on reply and full length blot should be added to the supplementary material as previously requested. 

The authors have added molecular weights to all blots as requested. However, this highlights the Nrf-2 size they are looking at is 68Kda, where in reality the activated version is approximately 100-110Kda, the authors need to address whether they have looked at this molecular weight size and put full gels in the supplementary data. 

In the discussion the new section at the end describing limitations needs to be re-written as the English is not clear.

Author Response

Reviewer (#3)' Comments to Author:

The authors have addressed the majority of comments including adding additional data. However they have not addressed my first and most important issue with the paper which is lack of any mitochondrial biogenesis assay. As previously stated this would add real weight to their hypothesis. The authors have stated this cannot be performed and have added a limitations section to offset this lack of experimental input.

à Response: The authors appreciate the reviewer’s comments. .

 If the manuscript is to be accepted it still needs further information around which distribution tests were performed on the data and why certain statistical tests were performed. A Shapiro-Wilk test and Kolmogorov-Smirnov test can be performed on data sets of six. So I dont accept the authors current explanation of their use of non-parametric testing. Moreover, the authors state they use Mann-Whitney U test and Kruskal-Wallis tests, but every Figure legend states that Mann-Whitney U tests were used. They need to justify why they are using non-parametric t-tests and only comparing two groups instead of using Kruskal wallis and comparing all three groups.

à Response: Thank you very much for your comments. We performed Kolmogorov-Smirnov test to determine the normality of distribution and found normal distribution. Therefore, the means for the groups were compared by analysis of variance followed by Tukey’s multiple comparison. Authors corrected them in the revised manuscript and did not mention statistics in the figures.

 The authors state when describing their SIRT3 mRNA data that "changes were not found in SIRT3 mRNA quantitative real-time PCR analysis because one SIRT3 mRNA expression was abnormally higher in Nx group (Supplementary Figure 4)" This is true but there is also two distinct populations in their controls which needs to be commented on and an explanation offered.

à Response: The authors appreciate the reviewer’s comments. As the reviewer pointed out, the SIRT3 values were distinct in two rats of normal controls and one rat of Nx group. Further experiments are likely to be necessary for accuracy. However, authors couldn’t perform additional experiments because we used all the remaining tissues for the last mRNA check. We suspect that the tissue’s storage status may have affected the results. We are very sorry that we cannot offer adequate explanation. Therefore, we removed QPCR data of SIRT3 in the revised manuscript.

 The authors have altered the actin blots as requested however the original targets blots have remained the same. So where have these actin blots come from? This should be made clear to the Editor on reply and full length blot should be added to the supplementary material as previously requested.

à Response: The authors appreciate the reviewer’s comments. We are sorry that authors initially presented mismatched actin blot. We corrected these mismatches in the 1st revised manuscript. We managed the first membrane for target protein detection without secondary antibody for beta actin (42 kDa) detection because we tried to detect target proteins near 42 kDa area. And then we simultaneously managed the second membrane for beta actin detection. Authors displayed full length blots in the last area of answers as Figure 1 to 3.

The authors have added molecular weights to all blots as requested. However, this highlights the Nrf-2 size they are looking at is 68Kda, where in reality the activated version is approximately 100-110Kda, the authors need to address whether they have looked at this molecular weight size and put full gels in the supplementary data.

à Response: Thank you very much for your comments. Antibody against Nrf2 was purchased from Abcam (Cambridge, MA, USA). Authors added full gels of Nrf2 (Figure 2) and data sheet (Figure 4).

In the discussion the new section at the end describing limitations needs to be re-written as the English is not clear. This is a generally well thought out paper based on good hypothesis it is just missing key mitochondrial biogenesis data, which would significantly add to the impact of the paper and would test the authors hypothesis further.

à Response: Thank you very much for your comments. Authors corrected it in the revised manuscript.

Round 3

Reviewer 1 Report

The authors did not perform necessary additional experiments, which I strongly recommended

Author Response

The authors did not perform necessary additional experiments, which I strongly recommended

Response: We are very sorry that we cannot perform additional experiments, which reviewer strongly recommended.

Reviewer 3 Report

I would except (even the actin response just!) all the authors responses aside from the following

1. Nrf-2. This authors/editorial team should read this paper https://www.ncbi.nlm.nih.gov/pmc/articles/PMC3503463/

I quote from the paper

"As new investigators break into the emerging field of Nrf2 research, confusion regarding the correct migratory pattern of Nrf2 is causing doubts about the accuracy and reproducibility of published results. This letter provides solid evidence that the biologically relevant molecular weight of Nrf2 is ∼95–110 kilodalton (kDa) and not the predicted ∼55–65 kDa based on its 2-kb open reading frame"

A recent paper in rats showed nrf-2 at approximately 100-110kda

https://www.ncbi.nlm.nih.gov/pmc/articles/PMC4922541/

when nrf-2 was knocked down the 110kda species was lost

The authors full length gels actually show higher bands that change (in nrf-1 and nrf-2) but as there are no molecular weight markers on there its impossible to say what size they are . The authors have this data and can perform densitometry and with this information in mind should readjust there analysis accordingly. If there are unsure they can run an assay with a known NRF2 inhibitor/activator as a positive control, to see which band size are altered.

2. The full length gels should be in the supplementary material with all the molecular weight markers on.

3. The SIRT 1 blot is not full length as its been cut at 100kDa right through the middle of another band in the controls that is not present in the Nx/Omega rats. What is this species? 

Author Response

I would except (even the actin response just!) all the authors responses aside from the following

  1. Nrf-2. This authors/editorial team should read this paper https://www.ncbi.nlm.nih.gov/pmc/articles/PMC3503463/

I quote from the paper

"As new investigators break into the emerging field of Nrf2 research, confusion regarding the correct migratory pattern of Nrf2 is causing doubts about the accuracy and reproducibility of published results. This letter provides solid evidence that the biologically relevant molecular weight of Nrf2 is 95–110 kilodalton (kDa) and not the predicted 55–65kDa based on its 2-kb open reading frame"

A recent paper in rats showed nrf-2 at approximately 100-110kda

https://www.ncbi.nlm.nih.gov/pmc/articles/PMC4922541/

when nrf-2 was knocked down the 110kda species was lost

The authors full length gels actually show higher bands that change (in nrf-1 and nrf-2) but as there are no molecular weight markers on there its impossible to say what size they are . The authors have this data and can perform densitometry and with this information in mind should readjust there analysis accordingly. If there are unsure they can run an assay with a known NRF2 inhibitor/activator as a positive control, to see which band size are altered.

Response: The authors appreciate the reviewer’s comments. We found bands which were suspected as Nrf2 (Supplementary Figure 8). We discussed it in the revised manuscript.

“The molecular weight of Nrf2 was found at 68  kilodalton (kDa) area in this study and other recent studies [23,24]. However, previous studies indicated biologically relevant molecular weight of Nrf2 is 95–110 kDa with evidence [25,26]. We also found bands which were suspected as biologically relevant Nrf2 (Supplementary Figure 8). Further studies are necessary to find molecular weight of biologically important Nrf2”.

  1. The full length gels should be in the supplementary material with all the molecular weight markers on.

Response: Thank you for your comments. We added full length gels in the supplementary figures (supplementary figure 2, 4, 6, 8).

  1. The SIRT 1 blot is not full length as its been cut at 100kDa right through the middle of another band in the controls that is not present in the Nx/Omega rats. What is this species?

Response: The authors appreciate the reviewer’s comments. We added full length gel of SIRT 1 in the supplementary figure 6.   

Round 4

Reviewer 3 Report

No further comments